# Relaxation and revival of quasiparticles injected in an interacting quantum Hall liquid

R. H. Rodriguez[1], F. D. Parmentier [1✉], D. Ferraro[2,3], P. Roulleau[1], U. Gennser [4], A. Cavanna [4], M. Sassetti[2,3], F. Portier[1], D. Mailly[4] & P. Roche[1]

The one-dimensional, chiral edge channels of the quantum Hall effect are a promising platform in which to implement electron quantum optics experiments; however, Coulomb interactions between edge channels are a major source of decoherence and energy relaxation. It is therefore of large interest to understand the range and limitations of the simple quantum electron optics picture. Here we confirm experimentally for the first time the predicted relaxation and revival of electrons injected at finite energy into an edge channel. The observed decay of the injected electrons is reproduced theoretically within a Tomonaga-Luttinger liquid framework, including an important dissipation towards external degrees of freedom. This gives us a quantitative empirical understanding of the strength of the interaction and the dissipation.

[1] Université Paris-Saclay, CEA, CNRS, SPEC, Gif-sur-Yvette 91191, France. [2] Dipartimento di Fisica, Università di Genova, Via Dodecaneso 33, 16146 Genova, Italy. [3] SPIN-CNR, Via Dodecaneso 33, 16146 Genova, Italy. [4] Université Paris-Saclay, CNRS, Centre de Nanosciences et de Nanotechnologies (C2N), Palaiseau 91120, France. ✉email: francois.parmentier@cea.fr

 

Electron quantum optics[1,2] is based on the profound analogy between the transport of single quasiparticles in a quantum coherent conductor and the propagation of single photons in a quantum optics setup. This has led to seminal electron interferometry experiments realized in edge channels (ECs) of the quantum Hall effect, whether in a Mach–Zehnder geometry[3] or, recently, in a Hong–Ou–Mandel setup[4] where two single-charge excitations emitted at a well-defined energy collide on a quantum point contact, probing their indistinguishable nature. The majority of these experiments have been performed at filling factor $\nu = 2$ of the quantum Hall regime, where, for a given carrier density, the quantum Hall effect is the most stable. However, interactions between the two ECs of $\nu = 2$ have been shown to lead to decoherence as well as energy relaxation. The latter corresponds to the fact that energy can be transferred from one EC to the next, even in the absence of tunneling between the two. This strongly challenges the simple picture of electron quantum optics, and raises the crucial question of the nature of the excitations that actually are interfering in the aforementioned experiments.

The first investigations of decoherence and energy relaxation at $\nu = 2$ involved biased quantum point contacts to generate a broadband, out-of-equilibrium distribution function that was probed using Mach–Zehnder interferometry[5–9] and energy spectroscopy[10,11] techniques. From these works emerged a clearer picture of the role of interactions between copropagating ECs, which is well accounted for by a powerful theoretical description in terms of Tomonaga–Luttinger liquid (TLL) physics. In the so-called TLL model, interactions lead to new eigenstates of the system, which are not Fermionic, but charge- and dipole- (or spin-) like plasmons shared by the two ECs[12–15]. The decomposition of a Fermionic excitation in one EC onto the plasmon modes shared by the two ECs gives rise to energy relaxation and decoherence[14–16]. This model describes particularly well Hong–Ou–Mandel collision experiments using single excitations emitted at finite energy[4,17–19].

Underlying the TLL model is the assumption that the interaction between the two ECs dwarfs all other energies. This means that although the energy of a carrier injected into one of the ECs will be redistributed between the two interacting ECs, the system will conserve its total energy. How valid this assumption is remains an important question, as a number of the basic predicted features of the evolution of a quasiparticle emitted at finite energy remain to be confirmed experimentally. The shape of the energy distribution of finite energy quasiparticles, which is referred to as the quasiparticle peak, has so far not been observed in the quantum Hall regime, nor has its evolution during propagation.

In fact, probing the quasiparticle peak is of crucial importance, since it would directly reflect the wavepackets of single particles that are manipulated in quantum optics, and its behavior could establish unambiguously characteristics specific to the TLL model. One such potential feature is the remarkable ability to partially regenerate the initial excitation[16]. This is analogous to Rabi oscillations, where a system oscillates between two states that are not proper eigenstates due to their mutual interaction. Specifically, the TLL predicted regeneration of an initial excitation comes about through the "catching up" and recombination of a fast-propagating charge plasmon with a slower dipole plasmon (animations illustrating the effect can be found in the Supplementary material of ref. [16]). However, this resurgence has only been indirectly observed in Mach–Zehnder interferometry experiments with biased quantum point contacts[5,7,20], whereas it should clearly appear as a revival of the quasiparticle peak at finite length and energy[16]. Furthermore, recent experiments using such finite energy

quasiparticles in other schemes revealed important qualitative inconsistencies with the TLL model. First, spectroscopy experiments showed that a sizable portion of the energy injected in the system was lost to additional degrees of freedom, not included in the TLL model[11]. Second, finite energy excitations were shown to interfere within a Mach–Zehnder setup with a visibility that decreased, but remained finite even at high energy instead of fully vanishing as predicted[21]. Very recently, an experiment using an energy spectroscopy technique similar to the one reported in the present paper showed that quasiparticles can exchange energy between spatially distinct parts of the circuit[22,23]. While this result can explain the missing energy reported in ref. [11], it is again in contrast with the TLL model. This series of inconsistencies raises a crucial question: is there merely a missing ingredient in the TLL model for it to fully describe the physics of interacting ECs, or is it necessary to replace it with a different theory? Indeed, a recent competing theoretical description[24] is qualitatively compatible with the early energy spectroscopy experiments[10,11]. Based on a Fermi liquid description of the ECs, and the assumption that electron–electron interactions do not conserve momentum, this model predicts that the quasiparticle peak gradually broadens and shifts towards lower energies while both ECs are warmed up. Contrary to double-step distribution functions obtained with a biased quantum point contact, which yield similar results within both models, the predicted behavior of finite energy quasiparticles is thus strikingly different, as the TLL model predicts the quasiparticle peak to diminish in amplitude, and then to revive, while its position and width remain constant.

To answer the above question, we have performed an experimental investigation of the energy relaxation of energy-resolved quasiparticles, showing a clear observation of the quasiparticle peak at $\nu = 2$. We show that while the quasiparticle peak is strongly suppressed with the injection energy and the propagation length, it clearly undergoes a revival at intermediate energy and length before disappearing into a long-lived state that is not fully thermalized. The observed evolution of the quasiparticle peak allows us to unambiguously discriminate between the two models. We show that the TLL model can be refined in order to explain our results by including dissipation towards external degrees of freedom, and, by spatially separating the two ECs with an additional gate, we unambiguously demonstrate the role of EC coupling.

## Results

**Experimental approach.** We have followed the approach proposed in refs. [24,25], and recently applied in ref. [22], in which one injects quasiparticles at a well-defined energy into an EC using a first quantum dot (QD) in the sequential tunneling regime. The injected quasiparticles then propagate over a finite length $L$, after which we perform a spectroscopy of the energy distribution function $f(E)$ of the quasiparticles using a second downstream QD as energy filter. This spectroscopy technique combined with a quantum point contact to generate excitations was previously used in refs. [10,11,26,27]. A very similar setup was used to investigate charge transfer processes between distant QDs in the absence of a magnetic field[28]; furthermore, a recent spectroscopy experiment showed that at vastly higher energies (in the 0.1 eV range), electrons in an EC decay by coupling to optical phonons[29]. It is also worth noting that other experimental techniques can be used to probe the energy distribution function, by measuring shot noise[30], or by performing a quantum tomography of the excitation injected in the EC[31–34]. The latter is known for being among the most challenging experiments undertaken so far in electron quantum optics.

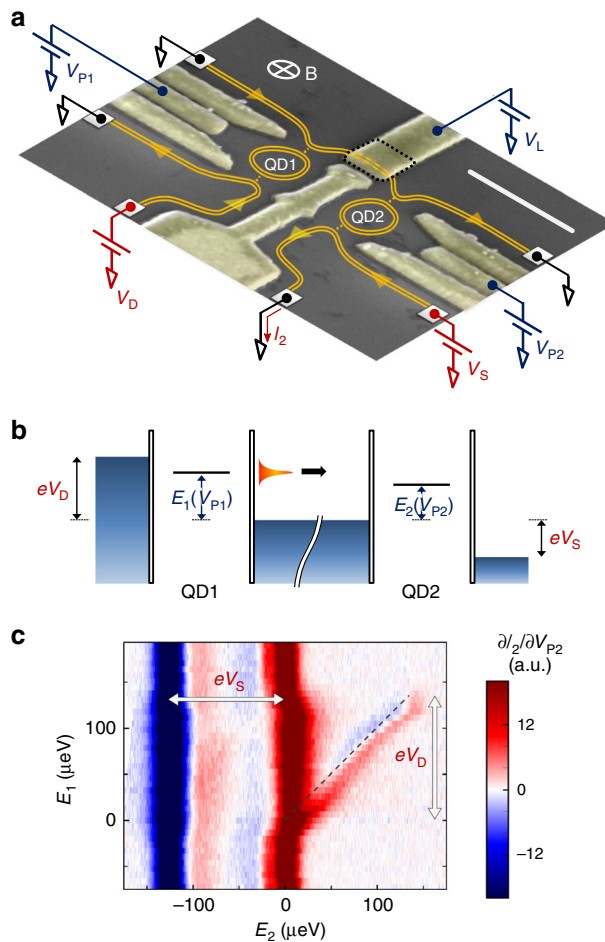

**Fig. 1 Principle and implementation of the experiment. a** False-color scanning electron micrograph of a typical sample. The ECs at $\nu = 2$ are depicted in orange. The large ohmic contacts located away from the center of the sample are depicted by the gray squares. The white scale bar corresponds to 500 nm. **b** Energy-scale sketch of the experiment. The two QDs are depicted by a single resonance at energy $E_1(V_{P1})$ and $E_2(V_{P2})$, respectively. The emitted quasiparticles are depicted by the orange bell-shaped curve. **c** Raw transconductance $\partial I_2/\partial V_{P2}$ of the second QD measured as function of $E_2(V_{P2})$ (x-axis) and $E_1(V_{P1})$ (y-axis). The thick vertical (resp. horizontal) arrow indicates the span of the drain (resp. source) potential $V_D$ (resp. $V_S$). The $y = x$ dashed line is a guide for the eye.

The devices' geometry is depicted in Fig. 1a. The two chiral ECs of $\nu = 2$ are depicted as orange lines. The QDs are defined electrostatically, and can be independently controlled using the plunger gate voltages $V_{P1}$ and $V_{P2}$. Both QDs are tuned to transmit only the outer EC. Quasiparticles in the outer EC stemming from the drain electrode are thus transmitted across the first dot QD1, and propagate along the outer EC connecting the first dot to the second dot QD2. A length gate, controlled by the voltage $V_L$, is used to increase the propagation path by diverting the ECs around the square area delimited by black dashed lines in Fig. 1a (a 200 nm insulating layer of SU-8 resist separates the rest of the gate from the surface of the sample). Several samples have been measured; here we show results obtained on three different devices, with nominal propagation lengths $L = 480$ nm, $L = 750$ nm, and 3.4 µm. Using the length gate on the first two devices yields the additional lengths $L \approx 1.3$ µm (long path for the $L = 480$ nm device) and $L \approx 2.17$ µm (long path for the $L = 750$ nm device—see Supplementary Note 1 for details on the devices, including the estimation of the lengths).

Figure 1b depicts the energy configuration of the two dots: a negative voltage $V_D$ is applied to the drain contact while the contacts connected to the ECs flowing between the two dots are grounded, defining the zero of energy in our experiment. A narrow single resonance of QD1 is tuned inside the transport window at an energy $E_1(V_{P1})$, defining the quasiparticle injection energy. We measure the transconductance $\partial I_2/\partial V_{P2}$ of QD2 while sweeping the energy $E_2(V_{P2})$ of a narrow single resonance in this dot that defines the detection energy. A calibration of both QDs is performed to extract their respective lever arms, linking the plunger gates voltages $V_{Pi}$ to the energies $E_i$ (see Supplementary Note 2). This allows us, after compensating for the small crosstalks between the two plunger gates, to directly probe the dependence of $\partial I_2/\partial V_{P2}$ with the detection energy $E_2$ for different values of the injection energy $E_1$. This signal is proportional to $-\partial(\Delta f(E))/\partial E$, where $\Delta f(E) = f(E) - f_S(E)$ is the difference of the energy distribution functions on either side of QD2[10,11,26,27], convoluted with the lineshape of the resonance of QD2 ($f_S(E)$ is the distribution function of the source EC). This convolution mostly affects the width of the features in the transconductance (see Supplementary Note 3). In the following, all widths discussed are convoluted widths. We separate the two contributions of $f(E)$ and $f_S(E)$ by applying a positive voltage $V_S$ to the source contact. This is illustrated in Fig. 1c, which shows a typical measurement of $\partial I_2/\partial V_{P2}$ as a function of $E_1$ and $E_2$, for $L = 480$ nm. The source and drain potentials, shown as thick arrows in Fig. 1c, are set to $eV_D \approx -eV_S \approx 125$ µeV, with $e \approx -1.6 \times 10^{-19}$ C the electron charge. The three main features appearing on this map are (i) the blue (negative) vertical line at $E_2 = eV_S \approx -125$ µeV, corresponding to $\partial f_S(E)/\partial E$, (ii) the red (positive) vertical line at $E_2 \approx 0$ µeV, corresponding to the low-energy part of $\partial f(E)/\partial E$, and (iii) the oblique line following a $y = x$ line (black dashed line), corresponding to the emitted quasiparticles, which are detected after their propagation. Note that no signature of Auger-like processes[22] (which would appear as diagonal lines dispersing in a direction opposite to the black dashed line) has been identified in any of the transconductance maps we obtained. We integrate the transconductance so as to obtain the energy distribution function $f(E)$, which we discuss in the rest of this paper.

**Measured distribution functions.** Figure 2 shows measurements of $f(E)$ for $L = 480$ nm (top panel) and $L = 750$ nm (bottom panel). The injection energy $E_1$ is gradually increased from negative values (blue curves), where the resonance of QD1 is outside the bias window, to large positive values $E_1 > 100$ µeV (red curves), where we expect to detect quasiparticles at high energy. The measured $f(E)$ curves evolve from a Fermi function at low temperature (the apparent temperature is increased to ~40 mK by the convolution with the resonance of QD2, see Supplementary Note 3) to strongly out-of-equilibrium distribution functions showing a distinct quasiparticle peak at finite energy. This is particularly striking for the shortest distance (top panel), where the peak clearly appears even at the largest energy $E_1 = 173$ µeV (note that the peak was not observed in ref. [22], where the propagation length was ~1.5 µm). The peak position increases linearly with $E_1$, while its amplitude decreases. In contrast, for a path only 50% longer, the peak amplitude is strongly suppressed; however, after vanishing at $E_1 \sim 90$ µeV, it reappears as $E_1$ is further increased (see inset in the bottom panel of Fig. 2). The clear presence of a quasiparticle peak, its strong decay, and its subsequent revival at intermediate lengths are consistently observed in our experiment, and are the main results of this paper. In the following, we quantitatively analyze the measured $f(E)$, and compare our results with the leading theories.

**Quasiparticle peak analysis.** Figure 3 shows a semi-log scale plot of the data shown in Fig. 2, illustrating our analysis. For $L = 480$ nm (Fig. 3a), the quasiparticle peak is well fitted by a Lorentzian peak without any offset, shown as dashed black lines. Remarkably, the energy position $E_{\text{peak}}$ of the peak matches the injection energy $E_1$, and its full-width at half-maximum remains constant as $E_1$ is increased (see inset in Fig. 3a). This observation, which was consistent in all data where the quasiparticle peak is distinguishable, is in direct contradiction with the predictions of Lunde and Nigg[24], but in agreement with the TLL model. Furthermore, the semi-log scale shows that the maximum of the quasiparticle peak follows an exponential decay (gray dashed line) over more than an order of magnitude. For $L = 750$ nm (Fig. 3b), the peak is strongly suppressed. However, while the peak only shows up as a faint bump at low $E_1$ and has vanished for intermediate $E_1$, it appears clearly at large $E_1$, and can again be fitted by a Lorentzian with preserved width and position. In addition, the peak height increases with the injection energy, as seen in

Fig. 2. We observed the revival in several realizations of the experiment in the same $L = 750$ nm device, with different gating conditions, and during different cooldowns (note that while we did not observe the revival for $L = 480$ nm, we show below that it is expected to occur at significantly higher $E_1$, outside our spectroscopy range—see "Methods"). While those observations clearly are characteristic features of the TLL model, it is not the case for the apparent exponential decay of the peak at $L = 480$ nm. Another discrepancy is the fact that the low-energy part of distribution functions, away from the quasiparticle peak, seem to be (at least to some extent) independent of the injection energy, whereas it should become broader with increasing $E_1$. This strongly suggests that dissipation—that is, loss of energy towards other degrees of freedom than the plasmon modes—needs to be taken into account. The presence of dissipation was already identified in previous works[11,35], and particularly in ref. [22], where it manifested as long-distance Auger-like processes.

**Modeling dissipation in the TLL model.** A simple way to include dissipation in the TLL model (see Supplementary Note 8 for details of the model) consists in introducing an ad hoc linear friction term in the equations of motion for the bosonic fields describing the charge and dipole plasmon modes[13,14,16,36]. Because of interactions, assumed here to be short-ranged, these modes are shared by the two ECs, and their respective velocities $v_\rho$ (charge mode) and $v_\sigma$ (dipole mode) depend on the Fermi velocities $v_1$, $v_2$ in each EC in the absence of interactions, as well as on the coupling $u$ between the ECs. These parameters combine into an effective mixing angle $\theta$, defined as $\tan(2\theta) = 2u/(v_1 - v_2) = 2u/v_2(\alpha - 1)$, which is zero when the two ECs do not mix, and $\pi/4$ for maximal coupling. This reflects the fact that even if the interaction $u$ is small, the ECs can become maximally coupled if they propagate at exactly the same velocity. In this strong coupling limit, and in the absence of dissipation, the quasiparticle peak height is given by a characteristic squared Bessel function $J_0^2(2.5 \times E_1/E_0)$, with $E_0 = 5\hbar v_\rho v_\sigma/L(v_\rho - v_\sigma) \approx 5\hbar v_\sigma/L$[14,16]. Its oscillatory behavior corresponds to the revival phenomenon, with the first zero occurring at $E_0$. Tuning $\theta$ away from the strong coupling value modifies the Bessel function profile, leading to a lifting up of the zeros. When one includes dissipation, expressions for the quasiparticle peak height are modified, and acquire an exponentially decaying prefactor $\sim \exp(-E_1/E_\gamma) = \exp(-2\gamma_0 E_1 L/\hbar v_\rho)$, where $\gamma_0$ is the friction coefficient. Note that the model can be further refined by, for example, considering non-linear plasmon dispersion[37,38], or long-range interactions[35].

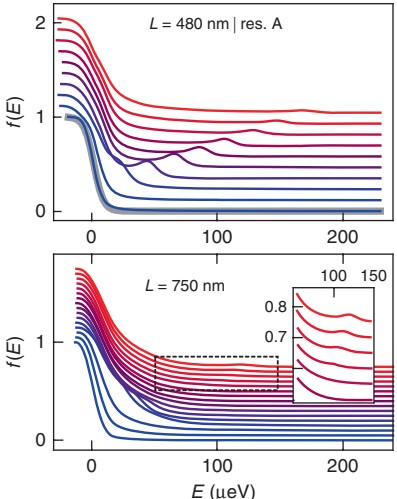

**Fig. 2 Measured distribution functions.** Top panel: measured $f(E)$ for $L = 480$ nm. Each curve, offset for clarity, corresponds to an increment of the injection energy $\delta E_1 \approx 21$ µeV, from $E_1 = -21$ µeV (blue) where no additional quasiparticles are emitted, to $E_1 = 173$ µeV (red). The thick gray line is a Fermi function fit of the data at $E_1 = -21$ µeV. Bottom panel: measured $f(E)$ for $L = 750$ nm. Each curve corresponds to a increment of the injection energy $\delta E_1 \approx 9$ µeV, up to $E_1 = 121$ µeV (red). The inset is a zoom on the region delimited by the dashed-line square. In all panels, the vertical offset is equal to $5.5 \times 10^3 \delta E_1$.

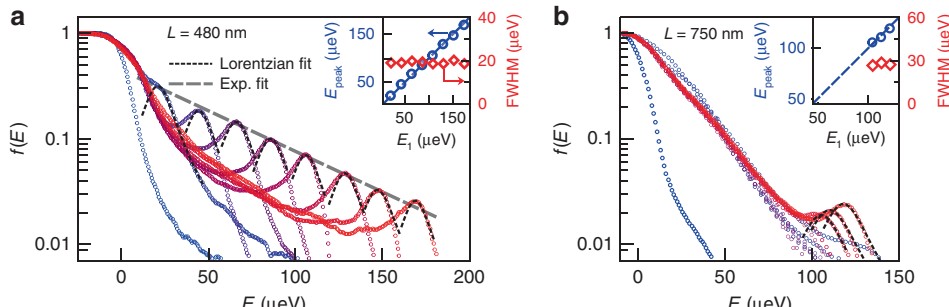

**Fig. 3 Exponential decay and revival of the quasiparticle peak.** Measured $f(E)$ for $L = 480$ nm (**a**) and $L = 750$ nm (**b**), in semi-log scale. The circles are experimental data, with the same dataset as in Fig. 2, ranging from $E_1 = \{-21, -17 \text{ µeV}\}$ (blue), to $E_1 = \{173, 121\}$ µeV (red). The black dashed lines are Lorentzian fits of the quasiparticle peak, and the gray dashed line in **a** is an exponential fit of the maxima of the Lorentzian fits, with a characteristic energy $E_d \approx 56$ µeV. Insets: Center $E_{\text{peak}}$ (blue circles, left $y$-axis) and full-width at half-maximum FWHM (red diamonds, right $y$-axis) of the Lorentzian fits, plotted versus injection energy $E_1$. The size of the symbols indicates our experimental accuracy. The blue dashed line in the insets is a $y = x$ line.

Figure 4 shows how this model compares to our data at $L =$ 480 and 750 nm. We plot the extracted Lorentzian peak heights from the 480 nm data shown in Fig. 3a (green pentagons), as well as for data obtained using a different resonance of QD1 in the same device (black hexagons), versus injection energy $E_1$. Data are normalized by the calibrated transmission of QD1, corresponding to the expected height of the injected peak (see Supplementary Note 1). The exponential decay observed in Fig. 3 is well reproduced by our model (thick green and gray lines). The TLL fits parameters $v_2$, $\alpha$, $\theta$, and $\gamma_0$, as well as corresponding plasmon velocities $v_\rho$, $v_\sigma$ and the characteristic energies $E_0$ and $E_\gamma$ are summed up in Table 1. In particular, the values of the revival energy $E_0$ obtained for the 480 nm sample are much larger than our maximum spectroscopy range ~ 200 μeV, explaining why the revival is not observed in that sample. We also plot in Fig. 4 the peak height for two different datasets of the $L =$ 750 nm device. The blue symbols (labeled cooldown 1) correspond to the data shown in Figs. 2 and 3b. The red symbols correspond to data obtained in a subsequent cooldown of the device, also showing the revival (see Supplementary Note 5 for additional data and analysis), despite having a different electrostatic environment due to thermal cycling. Importantly, this demonstrates that the observed revival is a robust phenomenon, unlikely to stem from a spurious mesoscopic effect (such as an impurity along the propagation path, or a parasitic resonance in one of the dots). In both datasets, the open symbols correspond to the peak height

extracted from the fits at large $E_1$, when the peak becomes visible again. The full symbols correspond to the value $f(E_1)$ of the measured distribution function taken at the injection energy. An important assumption here is that the peak position has not changed relative to the injection energy during propagation, which is validated for both $L$ by the Lorentzian fits. Again, our results are well reproduced by the model including dissipation (thick dark blue and dark red lines), particularly the observed revival, with parameters displayed in Table 1. Interestingly, because the exponentially decaying prefactor arising from the additional friction term in our model directly depends on the velocity of the charge mode $v_\rho$, we are able to extract all relevant parameters of the TLL model in our experiment. In contrast, the TLL analysis performed on most previous experiments[11,18,19,21,27,35] only provided the value of the dipole mode's velocity $v_\sigma$, while implying a strong coupling regime so that $v_\rho \gg v_\sigma$. Using a rather simple refinement of the TLL model, we are thus able to show that, in our experiment, (i) the Fermi velocities in the two ECs differ typically by a factor 2, (ii) the effective EC coupling is moderate, and that (iii) as a consequence, the difference between the charge and dipole plasmon velocities is not as large as usually assumed. Note that ref. [39] demonstrated that those velocities depend on the voltage applied to the gate defining the channel, reporting similar values (up to a factor 2) in our range of gate voltage. We also show that while the friction parameter is highly sample dependent, it does not depend on the QD resonances within a given sample, or on thermal cycling.

**Effect of the length gate**. Our analysis shows that the two devices differ not only by their nominal length and external dissipation but also by their plasmon velocities. Indeed, the 750 nm device presents a larger difference between $v_\rho$ and $v_\sigma$, effectively increasing the energy relaxation (or, in other words, making it effectively much longer than the 480 nm device as far as energy relaxation due to EC coupling is concerned). To interpolate between those two different cases, we rely on the length gate, the basic effect of which is illustrated in Fig. 5. For positive $V_L \approx 0.2$ V, the gate does not affect the trajectory of the ECs, which flow straight from QD1 to QD2 (Fig. 5a). The corresponding $f(E)$ measured for $L =$ 750 nm are shown in Fig. 5d, and are similar to the data shown in Fig. 2. For intermediate values $V_L \approx -0.1$ V, the electrostatic potential generated by the gate allows separating the two ECs[40], as depicted in Fig. 5b: spectacularly, in that case all data show a very clear quasi-particle peak up to large $E_1$ (Fig. 5e). In contrast, for large negative values $V_L \approx -0.5$ V, both ECs are diverted around the gate and follow a longer path ($L = 2.17$ μm, Fig. 5c), leading to the full disappearance of the quasiparticle peak even at low $E_1$ (Fig. 5f, see also Fig. 6). The quasiparticle peak evolution in the data shown in Fig. 5d, e can be reproduced using our model (see Supplementary Note 5), with slightly different Fermi velocities for the two datasets (but the same velocity ratio $\alpha = 2.1$). Interestingly, while the friction coefficient $\gamma_0 = 0.13$ is the same for the two datasets (as well as for the other measurements in

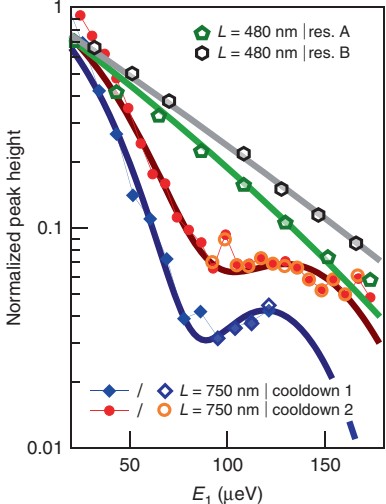

**Fig. 4 TLL fits with dissipation.** Normalized peak height plotted in semi-log scale as a function of injection energy $E_1$ for $L =$ 480 nm (green pentagons — resonance A, and black hexagons—resonance B) and $L =$ 750 nm (blue diamonds—cooldown 1, and red circles—cooldown 2). The size of the symbols corresponds to our typical experimental accuracy. The open symbols are the Lorentzian fits heights, and the full symbols the direct extractions of the amplitude $f(E_1)$ for $L =$ 750 nm. The thick lines are fits of the data using the TLL model including dissipation described in the text.

| Table 1 TLL fits parameters. | | | | | | | | |
|---|---|---|---|---|---|---|---|---|
| Sample | $v_2$ (km s$^{-1}$) | $v_\rho$ (km s$^{-1}$) | $v_\sigma$ (km s$^{-1}$) | $\alpha$ | $\theta$ | $\gamma_0$ | $E_0$ (μeV) | $E_\gamma$ (μeV) |
| 480 nm - res. A | 48 | 89 | 35 | 1.6 | 0.16 $\pi$ | 0.43 | 403 | 142 |
| 480 nm - res. B | 48 | 92 | 42 | 1.8 | 0.11 $\pi$ | 0.43 | 535 | 147 |
| 750 nm - cooldown 1 | 38 | 101 | 17 | 2.1 | 0.17 $\pi$ | 0.13 | 85 | 342 |
| 750 nm - cooldown 2 | 38 | 118 | 18 | 2.6 | 0.15 $\pi$ | 0.12 | 95 | 452 |

Lowest Fermi velocity $v_2$, charge and dipole plasmon velocities $v_\rho$ and $v_\sigma$, Fermi velocities ratio $\alpha$, effective inter-EC coupling $\theta$, friction coefficient $\gamma_0$, revival energy $E_0$, and exponential decay characteristic energy $E_\gamma$ extracted from the fits shown in Fig. 4.

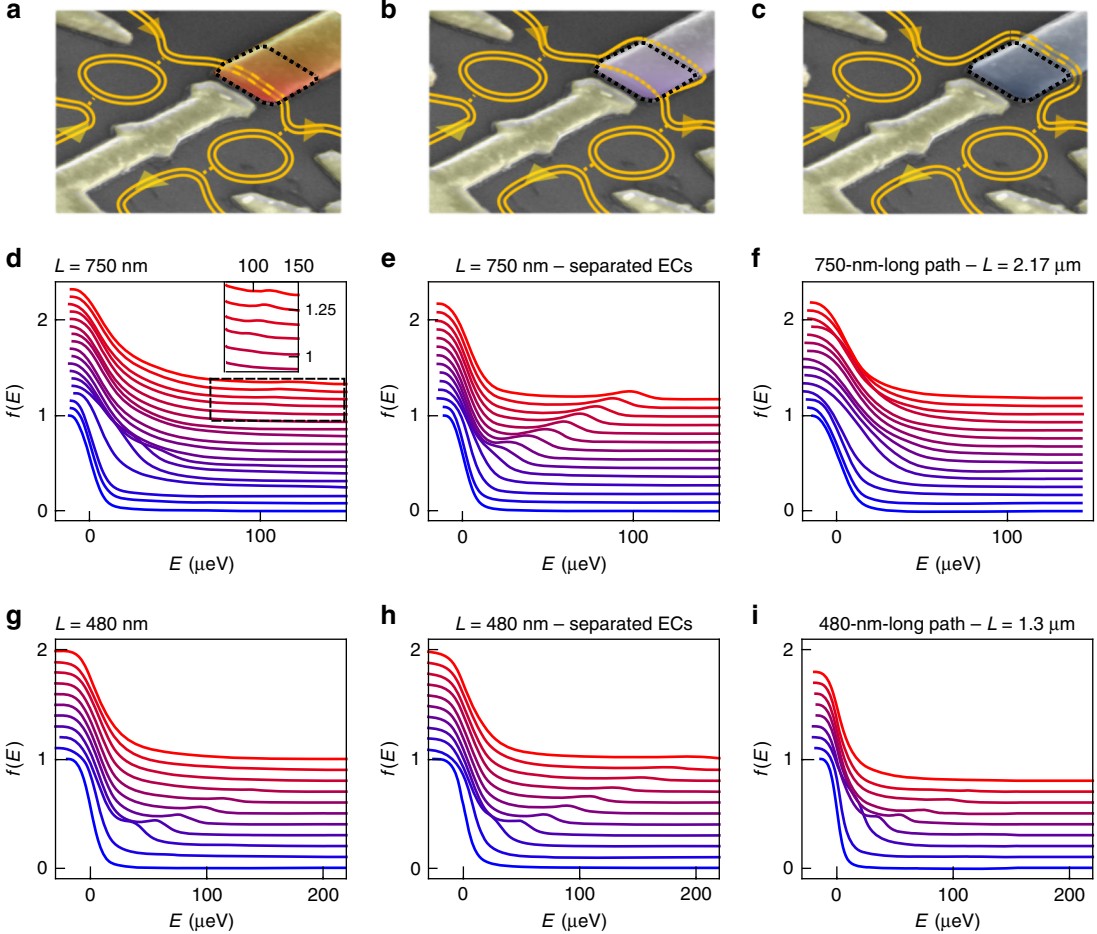

**Fig. 5 Effect of the length gate on energy relaxation. a–c** False-color scanning electron micrographs of a typical sample, depicting the trajectories of the ECs for $V_L = 0.2$ V (**a**, length gate highlighted in red), $V_L = -0.1$ V (**b**, gate highlighted in violet), and $V_L = -0.5$ V (**c**, gate highlighted in dark blue). In **a**, the ECs copropagate along the short path. In **b**, the ECs are spatially separated (orange dotted lines) as they flow below the length gate. In **c**, the ECs copropagate along the long path. **d–f** Measured $f(E)$ in the 750 nm device for the configurations depicted in respectively, **a**, **b**, and **c**. Each curve, offset for clarity, corresponds to an injection energy increment $\delta E_1 \approx 9$ μeV, from $E_1 \approx -26$ μeV (blue) to $E_1 \approx 122$ μeV (red, **d**)/$E_1 \approx 98$ μeV (red, **e**, **f**). The inset in **d** is a zoom on the region delimited by the black dotted square. **g–i** Measured $f(E)$ for the configurations in the 480 nm device depicted in respectively, **a**, **b**, and **c**. Each curve, offset for clarity, corresponds to an injection energy increment $\delta E_1 \approx 21$ μeV.

the $L = 750$ nm device), the extracted EC coupling $u \approx \{83, 21\}$ km s$^{-1}$ is four times smaller when the two channels are separated. The length gate on the 480 nm device makes it possible to manipulate the ECs in the same way (Fig. 5g–i), allowing us to separate the ECs (Fig. 5h), as well as to increase the copropagation length to $L \approx 1.3$ μm (Fig. 5i). For the latter length, the quasiparticle peak decreases sharply, but remains visible up to 100 μeV. The TLL analysis of both datasets shows that, as for the 750 nm device, the friction coefficient remains constant, $\gamma_0 = 0.43$ (see Supplementary Note 5 for additional plots and TLL analysis). For smaller gate voltages that do not fully separate the ECs, the length gate, coupled to the central gate separating the two QDs (see Fig. 1a), can nevertheless modify the electrostatic potential that defines the ECs flowing between the two dots, thereby granting us an additional control over the TLL parameters. We have performed the spectroscopy and TLL analysis of the quasiparticle peak height on the 480 and 750 nm devices for various gating configurations (see Supplementary Note 5 for plots and analysis, as well as a table summarizing the extracted TLL parameters). We observe consistently that the gate configuration allows tuning the plasmon velocities $v_\rho$ and $v_\sigma$, while the friction coefficient $\gamma_0$ remains constant in each device.

**Prethermalization**. We finally turn to the evolution of the measured distribution functions for propagation lengths above 1 μm. The integrability of the TLL model (in the absence of external dissipation) implies that energy relaxation should not lead to an equilibrium Fermionic state described by a high-temperature Fermi function[25,41]. This property has been recently confirmed by the observation of prethermalized states after the relaxation of highly imbalanced double-step distribution functions created by a biased quantum point contact[27], but, up to now, not for finite energy quasiparticles. We have observed that as the propagation length is further increased, the quasiparticle peak fully vanishes. Notably, when the peak is no longer visible, the distribution function does not qualitatively change, up to our longest studied length, $L = 3.4$ μm. This is illustrated in Fig. 6 (see also Fig. 5f), where we have plotted the measured $f(E)$ corresponding to the same injection energy $E_1 \approx 63$ μeV. Apart from the data at $L = 480$ nm, which display a clear quasiparticle peak, all other lengths yield similar, monotonous $f(E)$. These distribution functions cannot be fully fitted by a Fermi function: the dashed and dotted lines in Fig. 6 are tentative fits of the (respectively) low- and high-energy part of the distribution functions, with significantly different effective temperature for the high-energy part (~160–195 mK) with

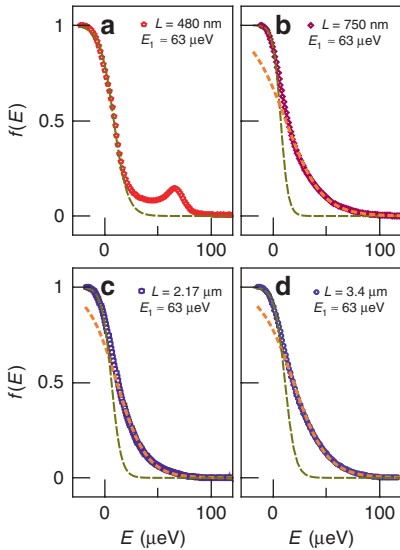

**Fig. 6 Length dependence and prethermalization.** Measured $f(E)$ for an injection energy $E_1 \approx 63\,\mu eV$ at **a** $L = 480$ nm, **b** $L = 750$ nm, **c** $L = 2.17\,\mu m$, and **d** $L = 3.4\,\mu m$. The symbols are experimental data, and the dotted (resp. dashed) lines are Fermi function fit of the high-energy (resp. low-energy) part of the data.

respect to the low-energy part (~40–55 mK—see also Supplementary Note 6). Both fits show significant deviations with respect to the data. As a sanity check, we have measured equilibrium distribution functions at elevated temperatures ($T \approx 160$ mK), corresponding to energy width similar to the data shown in Fig. 6, which showed much smaller deviations to a Fermi function (see Supplementary Note 6). Despite the significant role of energy losses towards external degrees of freedom, which should lead to a thermalized state after long propagation length, this apparent long-lived nonthermal behavior could indeed be a signature of TLL prethermalization. Furthermore, this might explain the recently reported robust quantum coherence of finite energy quasiparticles emitted in a Mach–Zehnder interferometer[21]. The apparent competition between prethermalization and observed dissipation is highly intriguing, and beckons further theoretical investigation of the impact of dissipation in the TLL model.

## Discussion

To summarize, we have directly observed the relaxation and revival of quasiparticles emitted at finite energy in an EC at filling factor $\nu = 2$ of the quantum Hall effect. These results qualitatively reproduces the hallmark phenomenology of the TLL model, and we show that the quantitative discrepancies are well accounted for by introducing dissipation in the model. In order to maximize the phase coherence and energy relaxation lengths in electron quantum optics experiments, one should not only rely on schemes that limit the effect of inter-EC coupling[26,42–44] but also identify the mechanisms behind this dissipation. A possible cause of this dissipation could be the recently observed long-distance Auger-like processes[22], although their signature is again not visible in our data. This stresses the need for further research in order to fully grasp the physics of interactions at $\nu = 2$[45].

## Methods

**Samples.** The samples were realized in a 90 nm-deep GaAs/GaAlAs two-dimension electron gas, with typical density $\sim 2.5 \times 10^{11}$ cm$^{-2}$ and mobility $\sim 2 \times 10^6$ cm$^2$ V$^{-1}$ s$^{-1}$, cooled down to electronic temperatures of $\sim 20$–$30$ mK. Perpendicular magnetic fields of about 5 T were applied to reach filling factor $\nu = 2$ of the quantum Hall effect.

**Measurements.** Measurements were performed in a dilution refrigerator, using standard low-frequency lock-in techniques. For each configuration of the experiment, the drain and source voltages $V_D$ and $V_S$ are tuned such that only a single narrow resonance sits in the transport window, with no excited states present. The spectroscopy range is then set by the minimum of $\{eV_D, |eV_S|\}$.

**Additional checks.** To ensure that no tunneling takes place between the two copropagating ECs, we check that the elevation of the electrochemical potential in the outer EC, obtained by integrating the measured $f(E)$, is equal to its expected value (see Supplementary Note 4).

## Data availability

The data and analysis used in this work are available from the corresponding author upon reasonable request.

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

## Acknowledgements

We warmly thank C. Altimiras, A. Anthore, G. Fève, F. Pierre, E. Bocquillon, X. Waintal, and P. Degiovanni for enlightening discussions, as well as P. Jacques for technical support. This work was funded by the CEA and the French RENATECH program.

## Author contributions

R.H.R. performed the experiments, with help from F.D.P., P.R., and P.Roc.; R.H.R., F.D.P., D.F., and P.Roc. analyzed the data, with inputs from U.G., D.M., and F.P.; D.F. and M.S. developed the theoretical model; U.G. and A.C. grew the 2DEG; D.M. fabricated the devices, with inputs from R.H.R., F.D.P., and P.Roc.; R.H.R., F.D.P., U.G., D.M., and P.Roc. wrote the manuscript, with inputs from all other coauthors; P.Roc. supervised the project.

## Competing Interests

The authors declare no competing interests.
