## [Peer Review File · Nature Communications]

Reviewers' comments:

Reviewer #1 (Remarks to the Author):

This paper reports a very interesting experiment, in which quasiparticle energy carried by charge and spin modes of a Luttinger liquid dephase and then rephase to produce a revival of the excitation energy a finite distance away in the Luttinger liquid.

My problem with the experiment is that the crucial aspects of the data come down to a comparison between two devices (480 and 750nm) that have very different behaviour, which is partly explained by the length but even moreso by the fact that extracted charge and spin velocities for the two make the 480nm device effectively much shorter and the 750nm device effectively much longer. There is no a priori reason that the charge and spin velocities should be different for the two (though I can imagine that accidental device details would change those velocities randomly), but it is an awfully convenient coincidence that the 480 and 750nm devices happen to be effectively so different. The result is that the one single device, the 750nm, is (for no very good reason) the only one out of many in which this effect is seen.

To sum up my overall objection, I would like to see the crossover from short to long behaviour not just in two accidental device configurations, and also see a few devices (not just a few cooldowns of the same device) that show the revival, probably at different energies.

As well, I have a few detail complaints:

1. The purpose of the V_L gate is not clear. In the first half of the paper the authors seem to suggest that this is used to tune from 480 to 750nm, though I understand from the rest of the paper that this is not what they mean to imply. My impression by the end is that each device can be operated in three configurations, tuned by V_L : short, long, and separated edge states. Only a few of the possible configurations (3 X number-of-devices) are described in the paper.
2. I don't understand from the SEM micrograph how the 750nm device is tuned to 2.15 μ m using V_L . Based in the picture I would expect 750nm1250nm or so, but not 2.15. Where does that number come from?
3. The description of the prethermalization part of the experiment is not clear to me. Why is the data fit with two Fermi functions? What is the meaning of what emerges from those fits? Why does TLL have to be invoked to explain the data, instead of simply an energy-dependent relaxation rate, and/or energy-dependent relaxation to non-edge-state degrees of freedom?

Reviewer #2 (Remarks to the Author):

In the manuscript, the authors report an energy spectroscopy experiment performed on copropagating edge channels of the quantum Hall state at filling factor $\nu = 2$. By injecting an electron into the outer edge channel through a quantum dot, an initial excitation having a well-defined energy is prepared in the copropagating channels. The excitation was detected after propagating about 1 μ m by another quantum dot working as an energy filter. The central results of the present manuscript are as follows:

1. A clear peak was observed in the shortest sample (480 nm) at low injection energy E_1 . This unambiguously indicates the detection of the initially prepared quasiparticle.
2. The peak height decreases exponentially with increasing E_1 , while the width of the peak is unchanged.
3. In the second shortest sample (750 nm), the corresponding peak is invisible at low E_1 whereas it becomes detectable at high E_1 .

The revival of the peak in the 750 nm device can be well understood by the Tomonaga-Luttinger

liquid (TLL) theory. On the other hand, decrease in the peak height with E_1 in the 480 nm device signals relaxation of quasiparticles in the edge channels, reflecting some dissipation mechanisms that are not included in the TLL model. The authors evaluated an empirical dissipation parameter as well as the TLL parameters from the experimental results, and confirmed consistency between data taken for several samples and different cooldowns using these parameters.

The experimental results and discussions are reasonable, and I think that clear observation of the quasiparticles' energy and its revival behavior in quantum Hall edge states should be interesting for broad readers in condensed matter physics. Although the origin of the dissipation remains unclear, the empirical parameters obtained in the present experiment are useful in this research field, as claimed by the authors. Therefore, I recommend this paper for publication in Nature Communications, after addressing following comments.

1. It is quite surprising that the quasiparticle peak becomes very small when the length of the channels becomes a little bit longer. How can I understand the length dependence? Is the length dependence specific for the present device? I could not understand why the long distance Auger-like processes can be the origin of the dissipation having such strong length dependence.

2. Tomography experiments have been performed not only for electrons in a 2-dimensional electron system [Jullien et al., Nature 514, 603 (2014)] but also for excitations propagating along quantum Hall edge channels [Bisognin et al, Nature Communications 10, 3379 (2019); Fletcher et al., Nature Communications 10, 5298 (2019)]. I think that it is worthwhile to mention these papers and explain the relation of the present research with them in a few sentences.

We thank both referees for their constructive remarks, which we address below. Please note that in the following, and similarly to our manuscript, we use the term *relaxation* to describe the energy redistribution between the edge channels, driven by the shared collective mode arising because of Coulomb interactions, and the term *dissipation* to describe energy losses towards external degrees of freedom.

Reviewer #1 (Remarks to the Author):

This paper reports a very interesting experiment, in which quasiparticle energy carried by charge and spin modes of a Luttinger liquid dephase and then rephase to produce a revival of the excitation energy a finite distance away in the Luttinger liquid.

My problem with the experiment is that the crucial aspects of the data come down to a comparison between two devices (480 and 750nm) that have very different behaviour, which is partly explained by the length but even moreso by the fact that extracted charge and spin velocities for the two make the 480nm device effectively much shorter and the 750nm device effectively much longer. There is no a priori reason that the charge and spin velocities should be different for the two (though I can imagine that accidental device details would change those velocities randomly), but it is an awfully convenient coincidence that the 480 and 750nm devices happen to be effectively so different. The result is that the one single device, the 750nm, is (for no very good reason) the only one out of many in which this effect is seen.

To sum up my overall objection, I would like to see the crossover from short to long behaviour not just in two accidental device configurations, and also see a few devices (not just a few cooldowns of the same device) that show the revival, probably at different energies.

We thank the referee for his/her positive comments and appreciation of the experiments' large importance. The main concern of the reviewer is that the revival effect is only seen in one device (the 750 nm one), and that we base our conclusions on experiments performed on three separate devices with different length (480 nm / 750 nm / 3.4 μm), and different behaviors. We understand this concern, and clearly, it would have been preferable to see the effect in a few devices at different energies. However, such additional experiments would hold up the diffusion of these observations for a long time. First, this would mean the fabrication of new samples. This process, though well defined, is never failsafe, and the time before having new samples should never be underestimated. The samples require two stable quantum dots showing narrow resonances and symmetric barriers. During our study we tested a number of additional devices where at least one of the quantum dots could not be operated in the required regime. Second, these samples would still need to be fully characterized at mK temperatures before the actual experiments could be started. Third, the parameter space being very large, the measurements and analysis in order to find a revival will certainly also be long. The revival is difficult to observe, as it requires the combination of several parameters which can become mutually exclusive, depending on the gating configuration. Indeed, the relaxation should be strong enough so that the revival occurs at an energy falling inside our typical measurement range (100-200 μeV), where none of the dots show excited states or additional distortions. However, if the relaxation becomes too strong, the quasiparticle peak fully vanishes and the revival is not observed (see e.g. the 2.17 μm data). Similarly, the dissipation should be weak enough so that it does not dominate.

The fabrication plus a full new measurement campaign would require at least several additional months, and probably an additional year, and therefore take too long with respect to the time allowed by the editors to provide a revised version.

We also believe that the submission of our work is very timely with respect to the development of electron quantum optics and decoherence and energy loss studies. We note that - as referee B quite rightly points out - the central results not only consist of the observed revival, but also of the first

experimental observation of the quasiparticle peak itself in a quantum Hall system (it was e.g. not observed in the recent work of Krähenmann et al., Nat. Commun. 2019), and of the quantitative characterisation of its decay (and thus a significant consolidation of the TLL model). We believe that the extended time for publication needed if we were to study new samples would prevent the healthy circulation of all these ideas within the community.

Nevertheless, we are confident that the triad of devices that we have investigated is sufficient to robustly establish our findings, as we here elucidate:

- 1) The revival is a robust phenomenon, observed for several gating conditions of the same device, in different cooldowns. Upon thermal cycling, the quantum dots are completely reset, as well as the local disorder configuration. This excludes any spurious mesoscopic effect (such as an impurity along the propagation path or a parasitic resonance in one of the dots) as the source of the revival. Importantly, it is highly reproducible (the distribution function data are obtained after averaging several plunger gate sweeps), and the fact that the revival can be suppressed or activated on a single sample by tuning the length with a gate, withdraws any other experimental artefact.
- 2) The main datasets shown in Fig. 2 of the manuscript show that, as the reviewer assesses, the 750 nm device is effectively much longer than the 480 nm device in terms of energy relaxation due to edge channel (EC) coupling, as usually described by the TLL model. However, the length gate allow us to tune not only the propagation, but also the TLL parameters. Combining this with the different dissipation rates shown by the 480 nm and 750 nm devices allows us to probe a sizeable part of the relaxation/dissipation parameter space.
- 3) The difference in TLL parameters is significant, but neither is it prohibitory large (the charge plasmons velocities differ by at most 40 %, and the spin velocities by a factor 2.5), nor is it unusual, as previous experiments reported sample-dependent velocities typically ranging from 27 to 87 km/s (Itoh et al., PRL 2018). The difference in external dissipation rate is more surprising, but this result might depend on the way this dissipation is introduced in the model. Indeed, in lack of a microscopic model, we have considered the simplest possible (linear) phenomenological way to include it, even if more involved dependences (i. e. quadratic) could be taken into account. Note that while both devices stem from the same 2DEG wafer, they have been fabricated two years apart (the 750 nm device was fabricated in 2016, and the 480 device in 2018), which might explain the different dissipation rates.

As well, I have a few detail complaints:

1. *The purpose of the V_L gate is not clear. In the first half of the paper the authors seem to suggest that this is used to tune from 480 to 750nm, though I understand from the rest of the paper that this is not what they mean to imply. My impression by the end is that each device can be operated in three configurations, tuned by V_L : short, long, and separated edge states. Only a few of the possible configurations (3 X number-of-devices) are described in the paper.*

Indeed, the 480 nm and 750 nm data are obtained on different samples, and the 3.4 μm data on a third sample with a different geometry. We have clarified our statement in the first half of the paper. As discussed above, the V_L gate present on the first two devices allows us to change the overall path length (by applying a large negative voltage so as to fully deplete the 2deg below the gate), as well as the inter-EC coupling (by applying smaller negative voltages on the gate so as to partially deplete the 2deg to a filling factor between 2 and 1). As stated by the reviewer, three main configurations can be obtained: short path (480 & 750 nm), long path (1.2 μm & 2.17 μm – see answer below for the estimation of the long path length), and separated ECs. In the first submitted version of the paper, we showed the three configurations for the 750 nm sample in the main text,

the short path configuration for the 480 nm sample in the main text, and the separated ECs configuration for the 480 nm sample in the supplementary information. In the new version, we have added the three configuration for the 480 nm sample in the main text. We also show several intermediary configurations of both devices, with their corresponding TLL analysis, in the supplementary information.

2. *I don't understand from the SEM micrograph how the 750nm device is tuned to 2.15 μ m using V_L . Based in the picture I would expect 750nm1250nm or so, but not 2.15. Where does that number come from?*

The SEM micrograph showing the resist bridge under the length gate correspond to a 480 nm sample. The length 2.17 μ m obtained for the long path of the 750 nm sample is extracted from its own micrograph, which we now show in the supplementary information.

3. *The description of the prethermalization part of the experiment is not clear to me. Why is the data fit with two Fermi functions? What is the meaning of what emerges from those fits? Why does TLL have to be invoked to explain the data, instead of simply an energy-dependent relaxation rate, and/or energy-dependent relaxation to non-edge-state degrees of freedom?*

We attempt to fit the data with several Fermi functions (at low and high energy) to illustrate that even though the distribution do not evolve in a qualitatively significant way after the disappearance of the QP peak, they do not correspond to thermal equilibrium states characterized by Fermi function. Furthermore, it highlights the fact that all data where the quasiparticle peak has vanished present similar deviations from a Fermi function. The fact that we observe a long-lived non-equilibrium state after the relaxation of the QP peak is highly reminiscent of the prethermalization phenomenon predicted by the TLL model. However, since there are no theoretical predictions so far for the energy distribution of a prethermalized state obtained by the relaxation of a quasiparticle in presence of external dissipation, we cannot rule out the fact that the observed distributions stem from another unidentified dissipation mechanism. We have clarified this in the new version of the paper.

Reviewer #2 (Remarks to the Author):

In the manuscript, the authors report an energy spectroscopy experiment performed on copropagating edge channels of the quantum Hall state at filling factor $\nu = 2$. By injecting an electron into the outer edge channel through a quantum dot, an initial excitation having a well-defined energy is prepared in the copropagating channels. The excitation was detected after propagating about 1 μ m by another quantum dot working as an energy filter. The central results of the present manuscript are as follows:

1. *A clear peak was observed in the shortest sample (480 nm) at low injection energy E_1 . This unambiguously indicates the detection of the initially prepared quasiparticle.*
2. *The peak height decreases exponentially with increasing E_1 , while the width of the peak is unchanged.*
3. *In the second shortest sample (750 nm), the corresponding peak is invisible at low E_1 whereas it becomes detectable at high E_1 .*

The revival of the peak in the 750 nm device can be well understood by the Tomonaga-Luttinger liquid (TLL) theory. On the other hand, decrease in the peak height with E_1 in the 480 nm device signals relaxation of quasiparticles in the edge channels, reflecting some dissipation mechanisms that are not included in the TLL model. The authors evaluated an empirical dissipation parameter as well as the TLL parameters from the experimental results,

and confirmed consistency between data taken for several samples and different cooldowns using these parameters.

The experimental results and discussions are reasonable, and I think that clear observation of the quasiparticles' energy and its revival behavior in quantum Hall edge states should be interesting for broad readers in condensed matter physics. Although the origin of the dissipation remains unclear, the empirical parameters obtained in the present experiment are useful in this research field, as claimed by the authors. Therefore, I recommend this paper for publication in Nature Communications, after addressing following comments.

- 1. It is quite surprising that the quasiparticle peak becomes very small when the length of the channels becomes a little bit longer. How can I understand the length dependence? Is the length dependence specific for the present device? I could not understand why the long distance Auger-like processes can be the origin of the dissipation having such strong length dependence.*

We thank the referee for his/her appreciation of the paper and its main results. We here address the specific comments. As discussed in the answer to reviewer #1's comment, the lengths 480 nm and 750 nm correspond to two different devices. The TLL analysis described in the paper indicates that the two devices have different characteristics, both in terms of plasmon velocities and of external dissipation rate (the 750 nm device shows a stronger EC coupling, while the 480 nm has more external dissipation). The former is not unusual, as previous experiments (including energy spectroscopy, or Hong-Ou-Mandel interferometry) have shown that the plasmon velocities can show significant variations from sample to sample. The latter is less clear; first because the dissipation rate difference is large (we find a factor 3 between the 480 nm and the 750 nm samples), and second because the dissipation mechanism is not identified. As we state above, maybe a different ad hoc description of this dissipation (for instance, a quadratic friction instead of a linear one) could yield more comparable rates between the two devices. Importantly though, when we use the length gate to either change the propagation length, or separate the ECs within a given device, our TLL analysis yields correspondingly changed length / inter EC coupling, while the dissipation rate remains unchanged. This consistency of parameters within a given devices gives us confidence in our analysis. To clarify this point, which is central to our paper, we now emphasize the fact the 480 nm and 750 nm data stem from two separate devices, and discuss in more depth the implications of the parameters extracted from our analysis.

Our results show that there is indeed a strong dissipation towards external degrees of freedom. Other previous experiments had already shown that dissipation is present in the system (le Sueur et al., PRL 2010 ; Bocquillon et al., Nat. Commun. 2013). However, as in our work, the cause of this dissipation is unknown. The recent work of Kraehenmann et al. identified one mechanism responsible for dissipation, that is, long range energy transfers. It is thus natural to wonder whether this mechanism is at work in our experiments. However, as we do not observe the experimental signatures of this mechanism (which would appear as oppositely dispersing diagonal lines in the transconductance map shown in Fig. 1 of our manuscript), and in absence of a full model encompassing TLL physics and Auger-like energy transfers, we can only speculate on the cause of the observed dissipation.

- 2. Tomography experiments have been performed not only for electrons in a 2-dimensional electron system [Jullien et al., Nature 514, 603 (2014)] but also for excitations propagating along quantum Hall edge channels [Bisognin et al, Nature Communications 10, 3379 (2019); Fletcher et al., Nature Communications 10, 5298 (2019)]. I think that it is worthwhile to mention these papers and explain the relation of the present research with them in a few sentences.*

We now cite the tomography papers (as well as the theoretical proposal - Grenier et al., New J. Phys. 13, 093007 (2011)) and discuss the relation with our work in the new version of the manuscript. A full quantum tomography indeed yields the energy distribution, along with information of the coherence of the detected wavepackets. Note however that these are usually extremely difficult experiments, with a lower sensitivity than our spectroscopy technique.

REVIEWERS' COMMENTS:

Reviewer #1 (Remarks to the Author):

The authors have made some helpful improvements in the paper. They do not address my concern about scarcity of samples, to confirm the robustness of the effect, except to say that it would just take too long to do this. I leave it up to the editor to decide whether the additional challenge of doing experimental double-checks outweighs the value of having a faster publication.

My personal judgement would be that the authors' explanation for their observations is probably correct, but I would not be particularly surprised if there turns out to be another explanation once more devices are made and tests. The authors' reliance on multiple cooldowns as being additional effective devices is not very helpful, in my experience, as identical gate structure and dopant configurations typically leads to very similar mesoscopic effects upon subsequent cooldowns, including defect-based effects like resonances.

One small suggestion: I found the sentence in the original paper "This is analogous to Rabi oscillations, where a system oscillates between two states that are not proper eigenstates due to their mutual interaction." to be very helpful, but sadly it did not make it through to the second version. I would add it, or another version of it, back, and possibly even expand the qualitative discussion of the effect for the non-expert research to better appreciate these effects.

Reviewer #2 (Remarks to the Author):

I find that the authors answered both referees' comments and questions politely, and revised the manuscript properly.

Now I recommend this manuscript for publication in Nature Communications.

We reiterate our thanks to both reviewers for their constructive comments. We address their latest remarks below:

Reviewer #1 (Remarks to the Author):

The authors have made some helpful improvements in the paper. They do not address my concern about scarcity of samples, to confirm the robustness of the effect, except to say that it would just take too long to do this. I leave it up to the editor to decide whether the additional challenge of doing experimental double-checks outweighs the value of having a faster publication.

My personal judgement would be that the authors' explanation for their observations is probably correct, but I would not be particularly surprised if there turns out to be another explanation once more devices are made and tests. The authors' reliance on multiple cooldowns as being additional effective devices is not very helpful, in my experience, as identical gate structure and dopant configurations typically leads to very similar mesoscopic effects upon subsequent cooldowns, including defect-based effects like resonances.

Again, we understand the concerns of Reviewer #1. As stated by the reviewer, the robustness of the revival with successive cooldowns does not indicate that thermal cycling amounts to testing a completely different device (and indeed, the parameters of the TLL model, particularly the dissipation, do not change drastically with thermal cycling), but rather demonstrate that it is a genuine observation and not an experimental artifact.

One small suggestion: I found the sentence in the original paper "This is analogous to Rabi oscillations, where a system oscillates between two states that are not proper eigenstates due to their mutual interaction." to be very helpful, but sadly it did not make it through to the second version. I would add it, or another version of it, back, and possible even expand the qualitative discussion of the effect for the non-expert research to better appreciate these effects.

We had removed the analogy with Rabi oscillations from the resubmitted version because we were not sure of its usefulness. We are happy that Reviewer #1 found it helpful, and have thus put it back in the present version. We also expand on the description of the revival by directly pointing the reader to very helpful animations that can be found in a theoretical paper published by one of the coauthors: Ferraro et al., PRL 2014.

Reviewer #2 (Remarks to the Author):

I find that the authors answered both referees' comments and questions politely, and revised the manuscript properly.

Now I recommend this manuscript for publication in Nature Communications.

We thank Reviewer #2 for his appreciation of our manuscript.